# Primary health care preparedness to integrate diabetes care in Blantyre, Malawi: A mixed methods study

Chimwemwe K. Banda[1,2]*, Ndaziona P. K. Banda[3], Belinda T. Gombachika[4], Moffat J. Nyirenda[1,5,6], Mina C. Hosseinipour[6,7,8], Adamson S. Muula[1,6,9]

1 School of Global and Public Health, Kamuzu University of Health Sciences, Blantyre, Malawi, 2 Public Health Group, Malawi Liverpool Wellcome Program, Blantyre, Malawi, 3 Department of Medicine, Kamuzu University of Health Sciences, Blantyre, Malawi, 4 School of Nursing, Kamuzu University of Health Sciences, Blantyre, Malawi, 5 Uganda MRC/UVRI Research Unit, Entebbe, Uganda, 6 NCD-BRITE Consortium, Kamuzu University of Health Sciences, Blantyre, Malawi, 7 University of North Carolina at Chapel Hill School of Medicine, Chapel Hill NC, United States of America, 8 UNC Project Malawi, Lilongwe, Malawi, 9 Africa Center of Excellence in Public Health and Herbal Medicine, Kamuzu University of Health Sciences, Blantyre, Malawi

* cbanda@cartafrica.org

**Data Availability Statement:** The datasets used and/or analysed during the current study are available from https://figshare.com/articles/dataset/

## Abstract

### Background

There is limited access to diabetes care services at primary care facilities in Malawi. Assessing the capacity of facilities to provide diabetes care is an initial step to integrating services at primary care.

### Aim

To assess the preparedness for delivering diabetes services at primary care level within the Blantyre District Health Office (DHO) to support the response to NCD epidemic in Malawi.

### Setting

Blantyre DHO primary care facilities.

### Materials and methods

A mixed methods approach nested in a national needs assessment for NCD response in Malawi was used. Fourteen primary healthcare facilities from Blantyre DHO were assessed. A tool adapted from the WHO rapid assessment questionnaire was used to identify human resource, equipment, supplies, and medication needed for comprehensive diabetes care. Descriptive statistics were done to analyze the quantitative data. Fisher's exact test was used to assess if there was a statistically significant difference between urban and rural facilities. Seventeen health care workers from the selected facilities participated in key informant interviews. Framework analysis method guided the qualitative data analysis. The quantitative and qualitative data were merged and displayed jointly.

NCD_focused_implimentation_research_needs_
assessment_NCD_BRITE_xlsx/14540391

**Funding:** Research reported in this publication was supported by the National Heart, Lung, And Blood Institute of the National Institutes of Health under Award Number U24HL136791. The content is solely the responsibility of the authors and does not necessarily represent the official views of the National Institutes of Health. The research was also supported by the Consortium for Advanced Research Training in Africa (CARTA). CARTA is jointly led by the African Population and Health Research Center and the University of the Witwatersrand and funded by the Carnegie Corporation of New York (Grant No. G-19-57145), Sida (Grant No:54100113), Uppsala Monitoring Center, Norwegian Agency for Development Cooperation (Norad), and by the Wellcome Trust [reference no. 107768/Z/15/Z] and the UK Foreign, Commonwealth & Development Office, with support from the Developing Excellence in Leadership, Training and Science in Africa (DELTAS Africa) programme. The statements made and views expressed are solely the responsibility of the Fellow. For the purpose of open access, the author has applied a CC BY public copyright licence to any Author Accepted Manuscript version arising from this submission. There was no additional external funding received for this study. The funders did not play any role in the study design, data collection and analysis, decision to publish or preparation of the manuscript.

**Competing interests:** The authors have declared that no competing interests exist.

## Results

The quantitative assessment showed that none of the facilities assessed had capacity to provide all the interventions recommended by WHO for diabetes care at primary level. Eight (57%) of the facilities had the capacity to diagnose diabetes, monitor glucose, prevent limb amputations and manage hypoglycemia and hyperglycemia. Four themes emerged from the qualitative data: differences in level of preparedness and implementation of diabetes care; disparities in resources between urban and rural facilities; low utilization of diabetes services; and strategy and policy recommendations for improvement of diabetes care.

## Conclusion

Inadequate health financing resulted in significant disparities in the available resources between the rural and urban facilities to offer diabetes care services. There is need to develop national policies and guidelines for diabetes care to strengthen the capacity of primary care facilities to facilitate achievement of universal health coverage.

## Introduction

According to the Sustainable Development Goals (SDGs), countries are required to reduce the morbidity and mortality rates of four main non-communicable diseases (NCDs), including cardiovascular diseases (CVDs), diabetes mellitus (DM), chronic respiratory diseases (CRDs), and cancer (CA), by one-third in 2030 from the 2015 level [1]. The problem is expected to be more complex in resource limited low- and middle-income countries (LMICs), where health systems are faced with a dual burden of infectious diseases and rise in non-communicable chronic conditions [2]. With inadequate and uneven distribution of specialized care facilities, most patients with diabetes in LMICs depend on an already overburdened primary care level health facilities for their health care needs [3]. Furthermore, the traditional orientation of health systems towards infectious disease management, limited resources and fragmented primary care, pose an uphill task for the primary care physicians in managing patients with diabetes and comorbidities [4, 5].

Strengthening and orienting the healthcare system to address the prevention and control of Non-Communicable Diseases (NCDs) NCD and the underlying social determinants is one of the priority areas for achieving universal health coverage in Malawi [6]. In Malawi, diabetes mellitus is one of the NCDs that cause significant morbidity and mortality, yet access to preventive care and treatment is a challenge for many [7]. The primary healthcare system has a vital role in prevention and control of diabetes and other NCDs [8, 9].

Malawi adopted the World Health Organization (WHO) framework for strengthening equity and efficiency of primary healthcare for NCDs in low and middle-income countries (LMICs) and launched a national action plan for NCD prevention and management in 2013 [10]. However, availability of diabetes services at primary care facilities especially in rural areas continue to be inconsistent in the country [10–12]. In Blantyre district, many people living with diabetes attend the diabetes clinic at Queen Elizabeth Central hospital (QECH), a specialized public referral and teaching hospital in the southern region of Malawi [13]. This is despite the existence of primary care facilities managed by the Blantyre district health office (DHO) within the district. Centralizing diabetes services limits accessibility and compromises care due to large patient numbers against staff shortages, limited laboratory capacity, and diabetes

medication stock outs [14]. In our previous study, we found that self-management support for people with diabetes attending the QECH diabetes clinic was inadequate due to staff shortages and high client turnover at the facility [15]. These findings highlighted the need of increasing access to diabetes services through primary care health facilities.

The lack of baseline data on burden of diabetes and the capacity of facilities to provide diabetes services is one of the challenges to integration of diabetes services into primary care services in many LMICs like Malawi [14, 16]. Studies assessing the capacity of health systems to deliver diabetes care in other sub-Saharan countries (Ghana [17], Zambia [18], Tanzania [19], Democratic Republic of Congo [18]) revealed that lack of equipment and basic medicines was common, however, each country had its own strengths and specific needs. Therefore, locally generated research data was needed to guide policy makers and service providers in response to NCDs and to identify existing resources and service needs in managing diabetes at primary care level in Malawi [14]. Introducing diabetes service at the primary care level without understanding the burden of disease and the capacity of the facilities to render such services may be cost ineffective and a waste of resource. Conducting a needs assessments before integrating diabetes services into primary care is therefore essential in addressing the knowledge-practice gap that exists [17, 20, 21].

The purpose of the assessment was to determine the preparedness of the primary care facilities in Blantyre DHO to provide diabetes services to support the response to the NCD epidemic in the Malawian health sector. We sought to determine the burden of diabetes; the size and capacity of human resource to manage diabetes; and availability of basic equipment, medicine and guidelines for the diagnosis and treatment of diabetes.

## Materials and methods

### Study design

We conducted a mixed methods study (Fig 1) to understand the context through which primary care services are provided in Blantyre DHO primary care facilities. A concurrent triangulation mixed methods strategy was used to collect the data whereby qualitative and quantitative data techniques were used concurrently [22]. The mixed methods approach was used to maximize the strengths and to minimize the weaknesses inherent in each approach [22].

### Study context

This study was nested in a larger study conducting a national needs assessment for NCD response in Malawi [23]. An estimated 468,000 people live with diabetes in Malawi [24]. The national prevalence of diabetes among adults aged 20–79 years olds is estimated at 7.3% [24]. Although district specific prevalence data is not available, records show that the prevalence of diabetes is high in both urban and rural areas [25]. The study setting was Blantyre DHO primary care facilities. In Malawi, health centers provide primary care services including under five care, outpatient and maternity services. At the time of data collection, Blantyre DHO had 28 health facilities of which 19 were health centers and 9 were community dispensaries. Seven of the health centers are located in urban area and twelve are rural. Unlike other DHOs in Malawi, the Blantyre DHO has no secondary level facility. All clients needing further treatment at the Blantyre DHO primary care facilities are referred to QECH, a regional tertiary hospital. Staffing at primary care facilities mostly comprise non-physician providers (medical assistants, clinical technicians or clinical officers) and nursing and midwifery technicians or registered nurses [26, 27].

Fourteen primary care facilities (seven urban and seven rural), from Blantyre DHO were included in the assessment. These facilities included seven health centers from Blantyre urban,

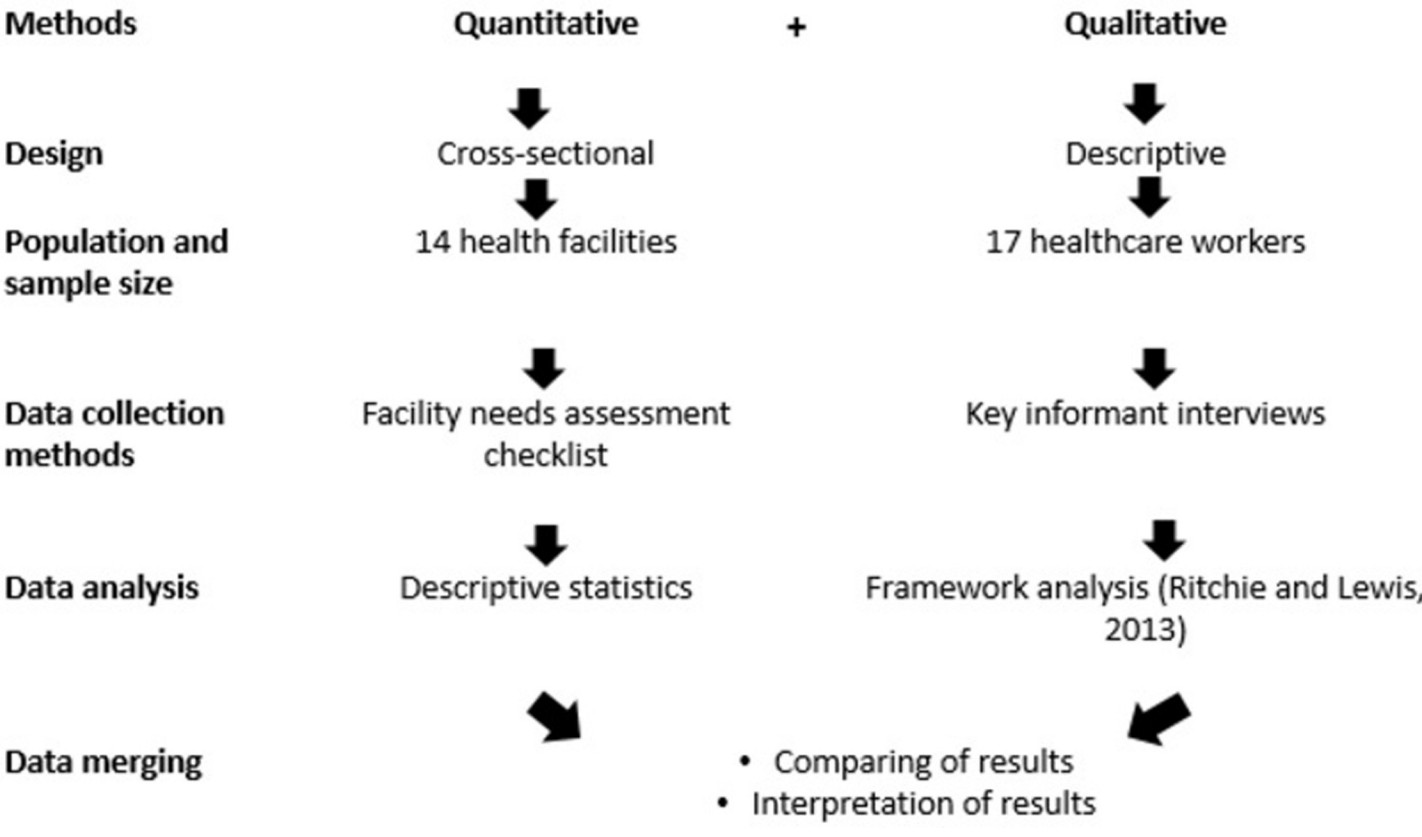

**Fig 1. Study design flow diagram.**

and seven rural health centers randomly selected and equaled with the number of the urban facilities. The fourteen facilities were considered enough to represent all the primary care facilities in assessing the capacity of Blantyre DHO to provide diabetes care services.

## Sampling

**Quantitative.** Fourteen facilities (seven urban and seven rural) from Blantyre DHO were included in the assessment. Given that urban Blantyre had only 7 facilities, we selected all sites to ensure good coverage [28]. We then randomly selected 7 health facilities in rural Blantyre to ensure a balance between rural and urban thus giving a total of 14 facilities. We did not perform a sample size calculation because of the small number of the sampling units (i.e. health centers). To randomly select the rural health centers, names of all the facilities were listed in an alphabetical order and numbered from one to twelve in Microsoft Excel. The RANDBETWEEN function in Microsoft excel was used to select the seven rural facilities to be assessed. The fourteen facilities were considered enough to represent all the primary care facilities in assessing the capacity of Blantyre DHO to provide diabetes care services

**Qualitative.** One or two healthcare workers were purposively selected from each participating health center to participate in key informant interviews. The participating healthcare workers included medical doctors, clinical officers, clinical technicians, medical assistants, nurses, pharmacy and physiotherapy departments' personnel. These people were recruited because of their knowledge and experience of working with people with diabetes or other NCDs at Primary care level for at least six months.

### Ethical consideration

Ethical clearance to conduct the study was granted by the College of Medicine Research Ethics Committee (COMREC) (Reference: P.07/19/2743). Permission to access the study sites was further sought from the Blantyre District Health Officer. Participation in the study was voluntary, and participants for the key informant interviews had to give a written consent. All identifiable data of participants were removed from the transcripts, and consent forms were stored in lockable cabinets where access was limited to the principal investigator to maintain participants' confidentiality.

### Data collection

All data collection was from 19[th] December 2019 and to 31[st] March 2020.

### Quantitative study

Quantitative data were collected using a needs assessment tool (S1 File.) which was developed to help assess the human and infrastructural capacity needs for NCD Response in health facilities in Malawi. The tool adapted the WHO questionnaire for the assessment of capacity to prevent and manage major NCDs in primary care centers in low-resource settings [29]. The tool has different sections to capture the following details: health facility profile; human resource and skills profile; facility-based diabetes prevalence; clinics and services; equipment; medicines and sundries; laboratory; costs related to diabetes; and referral system. For this present study, the assessment focused on diabetes services. Under each section, some items were added or removed from the original tool to make it comprehensive and context specific for the Malawi healthcare service.

The researcher or the research assistants administered the data tool to facility in-charges or their deputies and pharmacy personnel. Information on staff records for all facilities was collected from the Blantyre DHO human resource officer. Disease prevalence data for all facilities was obtained from the Blantyre DHO health management information system (HMIS) officers.

### Qualitative data

Sixteen key informant interviews were conducted with healthcare workers working at primary care facilities within Blantyre DHO. The first author or a research assistant, made appointments for interviews with the person in charge of the facilities or their representatives, in person or through a phone call. An information sheet was given to those who met the eligibility criteria (aged 18 and above, employed by ministry of health and having worked at the facility for at least six months) for them to read about the study. If they accepted to be interviewed, they were requested to sign the consent form. The research team member who obtained the consent also signed to confirm the person was informed about the study and was given an information sheet to read. The interviews were conducted on the day convenient to the participant in English as all the participants that were recruited were fluent in the language. A structured interview guide (S2 File.) adapted from the Malawi national needs assessment study was used for the key informant interviews [23]. To ensure alignment of the needs assessment with governments' health agenda, the sections of the interview guide were according to the following priority areas of the Malawi National Health plan: service delivery; preventive health and social determinants of health; leadership and governance; health financing; human resource for health; medicines, medical supplies, medical equipment and infrastructure; and health

information and research [6]. All interviews were recorded using a voice recorder and short notes were jotted during the interviews.

## Data analysis

**Quantitative data.** For the quantitative data, completed checklists and records were entered into a database created in Microsoft Excel. The data were exported into Stata Software version 14 for analysis. Descriptive statistics were calculated and presented as frequencies and percentages. Using the WHO guidelines for prevention and control of NCDs at primary care facilities in low resource settings, we considered individual facilities' capacity to provide diabetes services in six areas: (1) diabetes diagnosis; (2) glucose control; (3) reducing the risk of cardiovascular disease and diabetic nephropathy; (4) prevention of lower limb amputations; (5) prevention of blindness; and (6) management of severe hypoglycaemia and hyperglycaemic emergencies. Facilities were considered prepared if they had a) staff trained in diabetes management (during pre-service or in-service), b) necessary equipment and c) medications. Facilities were thus classified as prepared if they had all the three factors or not prepared if they had any of those factors missing. The differences in preparedness between urban and rural facilities was tested using the Fisher's exact test of association. A significance level of alpha equal to 0.05 was used to determine the statistical significance.

**Qualitative data.** Qualitative interview audios were transcribed verbatim by the first author, and where applicable they were translated from the vernacular language to English. Copies of transcripts were emailed to participants for them to comment if the transcript truly represents their views (member checking) [30]. The framework analysis method, according to Ritchie and Lewis guided the data analysis [31]. The first and second author read the transcripts to familiarize themselves with the data, then coded the data and agreed on the initial analytical framework. The initial analytical framework was based on the Malawi National Health Plan priority areas and the interview guide; and was applied to all the data. The coded data were then charted into corresponding cell of the framework. The analytical framework was further refined in an iterative process as new themes emerged from the data. The final thematic framework had four main themes.

## Results

Qualitative findings and quantitative results were integrated by merging to support each other [22]. The merged data are jointly displayed [32] in tables to show areas of convergence based on the qualitative themes. Four major themes emerged from the qualitative data: differences in level of preparedness and implementation of diabetes care; lack of resources; low utilization of diabetes services; and strategy and policy recommendations for improvement of diabetes care.

Table 1 shows the characteristics of participants in the qualitative interviews with staff from the 14 facilities that were included in the assessment.

### Theme 1: Differences in capacity and implementation of diabetes care

Six of the fourteen facilities that were assessed had a diabetes clinic. The actual services provided and capacity to offer recommended interventions differed among the facilities. None of the facilities had the capacity to implement all the WHO recommended interventions for diabetes care at primary care facilities (Table 2).

**Table 1. Qualitative interviews participants' characteristics.**

| Variable | Characteristic | n = 16 |
|---|---|---|
| **Sex** | Male | 5 |
| | Female | 11 |
| **Age** | 20–29 years | 10 |
| | 30–39 years | 6 |
| **Occupation** | Medical doctor | 6 |
| | Clinical Officer/ technician | 3 |
| | Medical Assistant | 5 |
| | Registered nurse/midwife | 1 |
| | Rehabilitation technician | 1 |
| **Work experience** | 1 – 5years | 11 |
| | 6–10 years | 4 |
| | 11–15 years | 1 |

## Theme 2: Lack of resources

Lack of adequate and appropriately trained human resources, medicines, infrastructure and diagnosis equipment was identified as a main problem at all facilities.

**Human resources.** Human resource was identified as one of the most vital needs to improve care. Fifty-two percent of the human resource available at the assessed facilities were Health Surveillance Assistants (HSAs). Other staff categories that were more common were nurse midwife technicians (24.15%) then medical assistants (10.1%). Only four (0.6%) of all the available staff reported to have attended a one-time training in NCD management. The key informants perceived that there was need for training and mentorship to equip those that lacked knowledge and skills in diabetes management.

*"We need the staff to be trained so they can be updated. Things are changing and we were trained way back."* **Participant8/medical assistant/male.**

*"Some of those cadres, they are not really trained, or they don't have the best experience when it comes to managing special cases or when say hypertension has complicated to heart failure. They are not the best people to actually manage those conditions. So we have no choice, but to*

**Table 2. Facilities with capacity or offering WHO recommended intervention for diabetes at primary level and source of guidelines.**

| Intervention | Facilities offering service (N = 14) | Facilities with capacity (N = 14) | Illustrative quotes |
|---|---|---|---|
| Diabetes diagnosis | 9 (64%) | 9 (64%) | *"We cannot diagnose diabetes. We don't have a glucometer."* Participant **12/medical assistant/male.** |
| Glucose control | 6 (43%) | 8 (57%) | *"For diagnosis, we are only able to really diagnose clinically... diabetes with the glucosticks and we have glucometers. For treatment, for diabetes we only have metformin and glibenclamide that we are able to give".* **Participant7/medical doctor/ female.** |
| Reducing the risk of cardiovascular disease and diabetic nephropathy | 3 (21%) | 3 (21%) | |
| Prevention of lower limb amputations | 6 (43%) | 8 (57%) | *"We have clinics in our district but most of our clinics [only] do drug refill."* **Participant1/medical doctor/female.** |
| Prevention of blindness | 2 (14%) | 2 (14%) | *"If we could have been doing the kidney function test. We check the urea, the creatinine, if our lab had those machines, at least we could decrease the burden at Queens."* **Participant4/medical doctor/male.** |
| Management of severe hypoglycaemia and hyperglycaemic emergencies | 9 (64%) | 9 (64%) | *"If we could have the ability to have frequent blood checks ...The other tests like the biochemistry tests, that would also be helpful ..."* **Participant7/medical doctor/female** |
| All interventions | 0 (0%) | 0 (0%) | |

*still have them on the rota because we don't have too many people. . ." **Partcipant7/medical doctor/female**.*

It was also reported that some of the practitioners lacked confidence to handle diabetes

*"In other facilities the services are there but the clinicians are afraid to take it up, I don't know why. . . so mostly they would just refer the patient to Queens or tell them to go to another health center where there is [diabetes] clinic".* **Participant2/medical doctor/female**

All facilities cited shortage of staff as a major challenge and the rural facilities were significantly more under-staffed compared to the urban facilities Table 3 below shows the median (interquartile range [IQR]) of different staff categories at the health facilities by location (urban/rural) and the recommended minimum numbers for nurse midwife technicians, medical assistant, and clinical technicians according to minimum standards of clinical services for Malawi [33].

**Medicines.** The diabetes medications stocked at the facilities were biguanide (Metformin) and sulphonylurea (Glibenclamide). Metformin was available at eight (57.14%) of the facilities, of which seven were urban and one was rural. All the facilities reported experiencing metformin stock out in the last quarter while five reported metformin stock out in the past year. Glibenclamide was available at seven (50%) of the facilities of which six were urban and one was rural. Two facilities reported stock out of glibenclamide in the last quarter and four facilities in the past year. Reasons for not stocking oral hypoglycemic medications were that the drugs are not supplied when ordered, or the clinicians were not confident to prescribe the drugs, or because the facility does not attend to many people with diabetes. All (100%) the facilities did not have any kind of insulin. The minimum standards for treatment of diabetes in Malawi recommend the availability of at least Metformin, Glibenclamide and Insulin at health centers

**Table 3. Number of different categories of human resources per facility by location.**

| Staff category | Urban Median (IQR) | Rural Median (IQR) | P value | Recommended numbers | Illustrative quote |
|---|---|---|---|---|---|
| Health Surveillance assistants | 37 (20–42) | 14 (8–20) | 0.0072 | 0* | *"We have a shortage of staff. For example, for us clinician, we are just two of us. That's a big challenge. Even the other cadres, like the HSAs are not enough. The main challenge is staffing. Like today, my colleague lost a relation and has gone out for a funeral. So I am the only one covering the facility for day and night shift."* **Participant13/medical assistant/female.** |
| Nurse midwife technicians | 15 (14–18) | 4 (3–11) | 0.0120 | 14 | |
| Medical Assistants | 5 (4–7) | 2 (1–3) | 0.0067 | 4 | |
| Community midwives | 1 (1–3) | 0 (0–1) | 0.0451 | 1 | *"I think the [human resource] size is small, very small against the patients that we receive. . . we have just one clinician who is doing all the screening, it affects the care that he is going to give because the same person is supposed to check the blood pressure, the same person is supposed to do the weighing scale and everything, so it's a little bit tiresome, and for example if I have fifteen or twenty patients, the way you are going to see those patients is very different than if you have a little more hands in.* |
| Registered nurse midwives | 4 (3–4) | 0 (0–3) | 0.0038 | 0* | |
| Clinical technicians | 2 (2–6) | 0 (0–2) | 0.0091 | 2 | |
| Laboratory technician | 1 (1–2) | 0 (0–0) | 0.0095 | 0* | |
| Medical doctor | 1 (1–1) | 0 (0–0) | 0.0005 | 0* | *"* **Participant5/clinical officer/ female"** |
| Clinical officers | 1 (0–2) | 0 (0–0) | 0.0245 | 0* | |
| Environmental Health | 0 (0–1) | 0 (0–0) | 0.0245 | 0* | |
| Laboratory assistant | 1 (0–1) | 0 (0–0) | 0.0241 | 0* | |
| Rehabilitation Assistant | 0 (0–0) | 0 (0–0) | 0.9165 | 0* | |

Key: * staff category not listed in the minimum standards of clinical services for Malawi

[33]. Table 4 below shows the number of urban and rural facilities that stocked the different classes of recommended diabetes medications.

Antihypertensives were available at most facilities. Hydrochlorothiazide was available at 13 (92.86%) facilities. Calcium channel blockers like nifedipine at 9 (64.29%) of the facilities. Beta blockers like propranolol were available at 6 (42.86%) of the facilities. ACE inhibitors were available at only 3 (21.43%) of the facilities. The antibiotics mostly available at the facilities were benzathine (92.86%), ceftriaxone (85.71%) and amoxillin (57.14%). For anticoagulants, warfarin and protamine were not available at any of the facilities while one (7.14%) facility had heparin. Other medications like statins were available at only one (7.14%) facility. Only five (35.71%) facilities reported that they receive the types of the medication they ask for, and two (14.29%) receive the quantities they order. Six (42.86%) had no refrigerator or air con in the pharmacy.

**Infrastructure and equipment.** None of the facilities had a specific room for diabetes care. At facilities which had diabetes service, the clinics were conducted at the out-patient department. A number of facilities did not have any of the essential basic equipment (Table 5).

## Theme 3: Low utilization of diabetes services

In facilities where diabetes services were available, utilization was low. Participants perceived that utilization was low due to lack of community awareness on the existence of the services and lack of trust in the services.

None of the facilities had community awareness campaigns, outreach or education programs for diabetes. The median (IQR) number of clients seen per facility due to diabetes in the previous twelve months was 174 (151–233) in the urban and 97 (42–110) in the rural (p = 0.0017). Key informants from urban facilities were of the view that many of the clients with diabetes were not utilizing the primary care facilities despite increase in diabetes cases within the catchment areas.

*"We don't see many cases of diabetes. . . most of them go to Queens [QECH]. . . ."* **Participant10/medical assistant/female**.

*"they are used to Queens [QECH] which is our main tertiary facility and referral facility, so these other clinics are not well patronized because our community outside there don't know that we have those services. . ."* **Participant1/medical doctor/female**.

inconsistencies in availability of diabetes medicines and supplies at the primary care facilities was the other reason cited for low client turnover. One participant had this to say:

*"We also have stock outs, that's often the main reason why people don't come. They say even if we go, we don't find the drugs."* **Participant6/medical doctor/male**.

**Table 4. Stocking of diabetes medication between urban and rural facilities.**

| Drug category | Urban | Rural | P value | Illustrative quote |
|---|---|---|---|---|
| | N = 7 | N = 7 | | |
| Biguanide | 7 (100%) | 1 (14%) | 0.002 | *"We don't have diabetes drugs. Because we don't order. We mostly don't know how to manage such cases."* **participant10/ medical assistant/female.** |
| Sulfonylurea | 6 (86%) | 1 (14%) | 0.015 | *"But the actual medicine like metformin or glibenclamide, we actually don't have. . . We don't stock that here. I'd say I haven't seen it in stock since I have been here. . .They don't supply.* **Participant11/clinical technician/male.** |
| Insulin | 0 (0%) | 0 (0%) | | *"Of course they don't give us insulin, insulin is given at Queens."* **Participant2/medical doctor/female.** |

**Table 5. Availability of essential basic equipment in urban and rural facilities.**

| Item | Urban Median (IQR) | Rural Median (IQR) | P value | Recommended numbers | Illustrative quote |
|---|---|---|---|---|---|
| Glucometers | 1 (1–1) | 0 (0–1) | 0.0072 | 1 | *"Yes, we have the glucometer but the sticks"* **participant1/medical doctor/female.** |
| Mercury BP† machines | 0 (0–1) | 0 (0–0) | 0.1422 | 1 | |
| Aneroid BP machines | 1 (0–2) | 0 (0–1) | 0.1967 | 1 | *"Another thing is to do with the BP cuffs, most times we run out of batteries. Sometimes we use a manual machine, but if using the manual one and you have so many patients, it takes time. It's better using the digital one . . . We are able to check for RBS but most times we run out of stock of the sticks, so it usually difficult to monitor."* **Participant11/medical assistant/male.** |
| Automatic BP Machines | 2 (1–4) | 1 (0–2) | 0.0654 | 1 | |
| Standard BP cuff | 2 (1–6) | 3 (2–4) | 0.6954 | 1 | *"We have BP cuff, however for most times, we have problems with batteries. So sometimes we use our own money to buy the batteries so that we can work. Other times the patients themselves when we tell them that we don't have batteries, some will say "but I want my BP to be checked, so I will go to buy the batteries". So that's the challenge we have with hypertension. While with diabetes, we don't have any equipment"* **Participaant13/medical assistant/female** |
| Alternate cuff | 0 (0–1) | 0 (0–0) | 0.4760 | 0* | |
| Bathroom scale | 1 (0–2) | 0 (0–1) | 0.0797 | 1 | |
| Hospital scale | 1 (1–3) | 1 (1–3) | 0.7362 | 1 | |
| Height meter | 0 (0–3) | 1 (0–1) | 0.7333 | 1 | |
| Ophthalmoscope | 0 (0–0) | 0 (0–0) | 0.3173 | 1 | |
| Snellen's chart | 0 (0–0) | 0 (0–0) | 0.3173 | 0* | |
| BMI‡ Chart | 1 (0–1) | 0 (0–1) | 0.4751 | 0* | |
| Monofilament | 0 (0–1) | 0 | | 0* | |

Key: † Blood Pressure

‡ Body Mass Index

* item not listed in the minimum standards of clinical services for Malawi

**Theme 4: Strategy and policy recommendations for improvement of diabetes care.** The participants suggested some strategies and policy recommendations to improve the primary health services for diabetes. Some suggested having national programs for NCDs same way it is for communicable diseases.

*"There should be national programs for NCD like it is with other diseases, for example end TB."* **Participant8/medical assistant/male**.

Other participants suggested screening services for NCDs at OPDs.

*"So I believe just to enhance on screening of NCDs, maybe if we would have rules established for basic things like vital signs for all the OPD patients so we can pick up patients that don't know. . . So enhancing primary screening to all OPD patients can be helpful."* **Partcipant3/ clinical officer/female**.

Another suggestion was on provision for management guidelines. The Malawi Standard Treatment Guidelines (MSTG) and the QECH medical department clinical book (referred to as the blue book) were available in 10 (71.43%) of the facilities. These guidelines were available as hard copies or electronic copies.

*"I think if we could have support with the guidelines. As I said most of the recommended guidelines that I know personally are the blue book, which is not really distributed to us."* **Participant7/medical doctor/female**

*"We have soft copies. Which we share on our phones. . . the one we receive is from Queens. We get them from Queens."* **Participant5/medical doctor/female**

These guidelines were not accessible to all service providers as one participant had this to say:

*"As for me, I have never seen any guidelines."* **Participant13/medical assistant/female**

The existing guidelines were however perceived as not offering adequate guidance to implementing diabetes services

*"Not all [diabetes] patients need to be seen at Queens or to be seen by doctors. Currently I know that some facilities are able to give a few of the medicine but they don't really offer reviews or basic checkups for those [diabetes] patients. So if that could be standardized to make sure it's happening, that would be good".* **Participant7/medical doctor/female**

## Discussion

The needs assessment suggested that primary care facilities within Blantyre DHO were not adequately prepared to integrate WHO guidelines for diabetes care [34]. Although some facilities were implementing some diabetes care, none of the facilities had the capacity to implement all interventions recommended by WHO (diabetes diagnosis; glucose control; reducing the risk of cardiovascular disease and diabetic nephropathy; prevention of lower limb amputations; prevention of blindness; and management of severe hypoglycaemia and hyperglycaemic emergencies) [8]. In particular, the assessment revealed that urban facilities were better prepared to carry out diabetes care services than rural facilities. For instance, the rural facilities lacked glucometers and diabetes medications which limited their capacity to diagnose diabetes and to offer glucose control interventions. Disparities in availability of resources between urban and rural facilities are common in most low and middle income countries [35]. The reason why urban facilities were better equipped for diabetes care could be the assumption that diabetes is disease of the affluent, therefore not considered an issue among rural communities. However, local data show that the burden of diabetes in rural and urban area of Malawi is similar [25]. Consequently, the lack of resources for diabetes care at rural facilities creates access barriers as people with diabetes are forced to travel to urban facilities for diabetes care. Our findings therefore highlight the need of strengthening both urban and rural primary care facilities to provide diabetes care.

Although all the urban facilities had at least one medical doctor, shortage of some basic equipment like blood pressure machines, testing reagents for glucometers or urine dipsticks for urinalysis and shortage of first-line drug regimens for hypertension and diabetes made it difficult for them to provide optimal care. While in rural facilities, the majority of the staff were non-physicians (medical assistants or clinical technicians) [26] who might have had no pre-service or in-service training in diabetes management. This made it difficult for them to know what equipment or medications to stock for diabetes management, or how to assist people with diabetes. If offered appropriate training and support, non-physician practitioners can be a useful resource in management of diabetes at primary care level [36]. Our findings on the limited capacity for primary care facilities to provide diabetes care in line with WHO recommendations concur with findings from other studies from within the region [17–19, 37]. Some have however argued that the lack of preparedness to manage diabetes at primary care facility is a reflection of non-optimal healthcare systems in low income countries [38].

Another reason for inadequate diabetes services at primary care level is the lack of national policy and guidelines for management of diabetes [39]. Only 53% and 59% of African and low income countries respectively have an operational policy, strategy or action plan for diabetes [40]. Malawi is one of the counties without a diabetes policy, strategy or action plan. Although the national health policy has a statement on improving diabetes care in the country [6], the national health sector strategic plan II categorizes diabetes treatment as unaffordable at the primary level therefore it is excluded from the national basic health package [41]. The available national prescription guidelines were not comprehensive enough to guide prevention, diagnosis, and management of diabetes, or its complications [42]. Similarly, the other available handbook which the clinicians consulted focused more on acute care for diabetes as it was intended for use at a tertiary facility [43]. Therefore, the need for having national guidelines or policy diabetes management was expressed by most of the healthcare workers who took part in the interviews. With sound policies, financing and guidelines, primary care facilities in Malawi have successfully responded to infectious condition like malaria [44], tuberculosis [45] and Human Immunodeficiency Virus (HIV) [46] despite resource limitations in the health sector. Therefore, the implication for policy is that to strengthen primary care facilities' capacity for diabetes care, there is need for national policy and guidelines for diabetes prevention and control.

Unfortunately, we also found that even where some diabetes care services were available, utilization of the services by surrounding communities was low. People with diabetes preferred to go to QECH, the regional referral hospital than their nearest primary care facilities. Local communities did not patronize the primary care facilities either due to lack of awareness on the existence of the services or due to lack of trust with in the primary care facilities to find the service they needed when they present there. Therefore, there is need to increase public awareness on the available diabetes services at primary care level and ensuring consistency of available services. A study assessing the feasibility of decentralizing diabetes care at primary healthcare facilities in Zomba, Phalombe, Machinga and Mulanje districts in Malawi, managed to retain 73% of people with diabetes in care over a 12 months period [7]. Staff trainings in diabetes care, mentorship programs, drug supply system strengthening, record keeping and peer education are some of the strategies that were used in this study [7]. Similar strategies could be adopted in Blantyre DHO primary care facilities to strengthen the capacity of diabetes care and maximize reach to people requiring the services within the catchment area of the facilities. Additionally, equipment for diabetes diagnosis and monitoring of glucose control and diabetes complications should be consistently made available at the primary care facilities.

The strength of our study was that by utilizing a mixed methods approach to the assessment, we were able to include narrative explanations to support the figures, and each method compensated the other methods' weaknesses. Secondly the setting of our study allowed us to compare rural and urban facilities unlike other DHOs in the country which only have rural facilities. There were however some limitations to our study. Firstly, we are not able to make statistical generalizations from the findings as data were collected at primary care facilities from one DHO. Secondly, we relied on the informants reports, therefore there was a possibility of social desirability bias, where participants present themselves in the most favorable manner [47].

## Conclusion

Integration of diabetes care at primary level facilitates the achievement of universal accesses to health care for all. However, shortage of skilled staff, medication and equipment limits the capacity of primary care facilities in Blantyre DHO to provide diabetes services. A national policy and guidelines for diabetes care are also needed in order to strengthen the capacity of primary care facilities. Given the right support, supervision and mentorship, the available staff

mix should be able to diagnose and treat diabetes at primary care level, and refer where necessary.

## Supporting information

**S1 File. Needs assessment tool.**
(DOCX)

**S2 File. In-depth interview guide for key informants.**
(DOCX)

## Acknowledgments

We would like to thank the Blantyre DHO management and staff for their support and assistance in the needs assessment.

## Author Contributions

**Conceptualization:** Chimwemwe K. Banda, Moffat J. Nyirenda, Mina C. Hosseinipour, Adamson S. Muula.

**Data curation:** Chimwemwe K. Banda.

**Formal analysis:** Chimwemwe K. Banda, Belinda T. Gombachika.

**Funding acquisition:** Moffat J. Nyirenda, Mina C. Hosseinipour, Adamson S. Muula.

**Investigation:** Chimwemwe K. Banda, Moffat J. Nyirenda, Adamson S. Muula.

**Methodology:** Chimwemwe K. Banda, Ndaziona P. K. Banda, Belinda T. Gombachika, Moffat J. Nyirenda, Mina C. Hosseinipour, Adamson S. Muula.

**Project administration:** Chimwemwe K. Banda.

**Resources:** Chimwemwe K. Banda, Ndaziona P. K. Banda, Moffat J. Nyirenda, Mina C. Hosseinipour, Adamson S. Muula.

**Supervision:** Belinda T. Gombachika, Moffat J. Nyirenda, Mina C. Hosseinipour, Adamson S. Muula.

**Validation:** Ndaziona P. K. Banda, Belinda T. Gombachika, Moffat J. Nyirenda, Mina C. Hosseinipour.

**Writing – original draft:** Chimwemwe K. Banda.

**Writing – review & editing:** Chimwemwe K. Banda, Ndaziona P. K. Banda, Belinda T. Gombachika, Moffat J. Nyirenda, Mina C. Hosseinipour, Adamson S. Muula.

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
