## [Decision Letter · Decision Letter 0]

24 Jan 2024

PONE-D-23-35256Primary healthcare preparedness to integrate diabetes care in Blantyre, Malawi: a mixed methods studyPLOS ONE

Dear Dr. Kwanjo Banda,

Thank you for submitting your manuscript to PLOS ONE. After careful consideration, we feel that it has merit but does not fully meet PLOS ONE’s publication criteria as it currently stands. Therefore, we invite you to submit a revised version of the manuscript that addresses the points raised during the review process.

We look forward to receiving your revised manuscript.

Kind regards,

Christine Kelly, PhD

Academic Editor

PLOS ONE

Journal Requirements:

Research reported in this publication was supported by the National Heart, Lung, And Blood Institute of the National Institutes of Health under Award Number U24HL136791. The content is solely the responsibility of the authors and does not necessarily represent the official views of the National Institutes of Health.

This research was also supported by the Consortium for Advanced Research Training in Africa (CARTA). CARTA is jointly led by the African Population and Health Research Center and the University of the Witwatersrand and funded by the Carnegie Corporation of New York (Grant No--B 8606.R02), Sida (Grant No:54100029), the DELTAS Africa Initiative (Grant No: 107768/Z/15/Z). The DELTAS Africa Initiative is an independent funding scheme of the African Academy of Sciences (AAS)’s Alliance for Accelerating Excellence in Science in Africa (AESA) and supported by the New Partnership for Africa’s Development Planning and Coordinating Agency (NEPAD Agency) with funding from the Wellcome Trust (UK) (Grant No: 107768/Z/15/Z) and the UK government, the statements made and views expressed are solely the responsibility of the fellow.

Research reported in this publication was supported by the National Heart, Lung, And Blood Institute of the National Institutes of Health under Award Number U24HL136791. The content is solely the responsibility of the authors and does not necessarily represent the official views of the National Institutes of Health.

This research was also supported by the Consortium for Advanced Research Training in Africa (CARTA). CARTA is jointly led by the African Population and Health Research Center and the University of the Witwatersrand and funded by the Carnegie Corporation of New York (Grant No--B 8606.R02), Sida (Grant No:54100029), the DELTAS Africa Initiative (Grant No: 107768/Z/15/Z). The DELTAS Africa Initiative is an independent funding scheme of the African Academy of Sciences (AAS)’s Alliance for Accelerating Excellence in Science in Africa (AESA) and supported by the New Partnership for Africa’s Development Planning and Coordinating Agency (NEPAD Agency) with funding from the Wellcome Trust (UK) (Grant No: 107768/Z/15/Z) and the UK government, the statements made and views expressed are solely the responsibility of the fellow.

Research reported in this publication was supported by the National Heart, Lung, And Blood Institute of the National Institutes of Health under Award Number U24HL136791. The content is solely the responsibility of the authors and does not necessarily represent the official views of the National Institutes of Health.

This research was also supported by the Consortium for Advanced Research Training in Africa (CARTA). CARTA is jointly led by the African Population and Health Research Center and the University of the Witwatersrand and funded by the Carnegie Corporation of New York (Grant No--B 8606.R02), Sida (Grant No:54100029), the DELTAS Africa Initiative (Grant No: 107768/Z/15/Z). The DELTAS Africa Initiative is an independent funding scheme of the African Academy of Sciences (AAS)’s Alliance for Accelerating Excellence in Science in Africa (AESA) and supported by the New Partnership for Africa’s Development Planning and Coordinating Agency (NEPAD Agency) with funding from the Wellcome Trust (UK) (Grant No: 107768/Z/15/Z) and the UK government, the statements made and views expressed are solely the responsibility of the fellow.

5. Please amend either the title on the online submission form (via Edit Submission) or the title in the manuscript so that they are identical.

6. Please upload a new copy of Figure 1 as the detail is not clear. Please follow the link for more information: https://blogs.plos.org/plos/2019/06/looking-good-tips-for-creating-your-plos-figures-graphics/" https://blogs.plos.org/plos/2019/06/looking-good-tips-for-creating-your-plos-figures-graphics/

Reviewers' comments:

Reviewer's Responses to Questions

**Comments to the Author**

1. Is the manuscript technically sound, and do the data support the conclusions?

Reviewer #1: Yes

2. Has the statistical analysis been performed appropriately and rigorously? 

Reviewer #1: Yes

3. Have the authors made all data underlying the findings in their manuscript fully available?

Reviewer #1: Yes

4. Is the manuscript presented in an intelligible fashion and written in standard English?

Reviewer #1: Yes

5. Review Comments to the Author

Reviewer #1: A good and informative paper. However, the authors need to give more details in the methodology and result sections. Specifically, there is no specific formula quoted in finding the sample size of 14 in the quantitative data collection. In the results section, there are two tables where it would be useful for the authors to provide national standard figures for staff and equipment. There are certain paragraphs in the the discussion that would be better off rephrased. Further, there's little discussion on the differences between the urban and rural facilities and how the differences impact utilization in the 2 areas. More details of my comments are provided in the paper against specific sections.

6. PLOS authors have the option to publish the peer review history of their article (what does this mean?). If published, this will include your full peer review and any attached files.

Reviewer #1: No

---

## [Author Response · Author response to Decision Letter 0]

7 Mar 2024

The funding statement has been updated and should read:

Research reported in this publication was supported by the National Heart, Lung, And Blood Institute of the National Institutes of Health under Award Number U24HL136791. The content is solely the responsibility of the authors and does not necessarily represent the official views of the National Institutes of Health.

CKB was supported by the Consortium for Advanced Research Training in Africa (CARTA). CARTA is jointly led by the African Population and Health Research Center and the University of the Witwatersrand and funded by the Carnegie Corporation of New York (Grant No. G-19-57145), Sida (Grant No:54100113), Uppsala Monitoring Center, Norwegian Agency for Development Cooperation (Norad), and by the Wellcome Trust [reference no. 107768/Z/15/Z] and the UK Foreign, Commonwealth & Development Office, with support from the Developing Excellence in Leadership, Training and Science in Africa (DELTAS Africa) programme. The statements made and views expressed are solely the responsibility of the Fellow. For the purpose of open access, the author has applied a CC BY public copyright licence to any Author Accepted Manuscript version arising from this submission."

There was no additional external funding received for this study.

---

## [Editor Report · Decision Letter 1]

16 Apr 2024

Primary healthcare preparedness to integrate diabetes care in Blantyre, Malawi: a mixed methods study

PONE-D-23-35256R1

Dear Dr. Banda,

We’re pleased to inform you that your manuscript has been judged scientifically suitable for publication and will be formally accepted for publication once it meets all outstanding technical requirements.

Kind regards,

Christine Kelly, PhD

Academic Editor

PLOS ONE
---

## [Editor Report · Acceptance letter]

2 May 2024

PONE-D-23-35256R1 

PLOS ONE

Dear Dr. K. Banda, 

I'm pleased to inform you that your manuscript has been deemed suitable for publication in PLOS ONE. Congratulations! Your manuscript is now being handed over to our production team.

Kind regards, 

on behalf of

Dr. Christine Kelly 

Academic Editor

PLOS ONE